# Does Digital Nature Enhance Social Aspirations? An Experimental Study

**DOI:** 10.3390/ijerph17041454

**Published:** 2020-02-24

**Authors:** Josca van Houwelingen-Snippe, Thomas J. L. van Rompay, Menno D. T. de Jong, Somaya Ben Allouch

**Affiliations:** 1Communication Science, University of Twente, De Zul 10, 7522 NJ Enschede, The Netherlands; t.j.l.vanrompay@utwente.nl (T.J.L.v.R.); m.d.t.dejong@utwente.nl (M.D.T.d.J.); 2Digital Life Centre, University of Applied Sciences Amsterdam, Wibautstraat 2, 1091 GM Amsterdam, The Netherlands; s.ben.allouch@hva.nl

**Keywords:** digital nature, experimental study, spaciousness, social aspirations, loneliness

## Abstract

Loneliness and social isolation are increasingly recognized as important challenges of our times. Inspired by research hinting at beneficial effects of interacting with nature on social connectedness and opportunities provided by ambient technology to simulate nature in a rich and engaging manner, this study explored to what extent digital nature projections can stimulate social aspirations and related emotions. To this end, participants (N = 96) were asked to watch, individually or in pairs, digital nature projections consisting of animated scenes which were either dense or spacious and depicting either wild or tended nature. Subsequently, they filled out a questionnaire comprising measures for social aspirations, awe and fascination. Results show that spacious scenes elicited significantly higher social aspiration and awe scores, especially when watching alone. Design implications are discussed for making digital nature accessible for people with limited access to real nature.

## 1. Introduction

Loneliness, both amongst young people and older adults, is increasingly recognized as a serious public health concern [1]. Apart from its negative effect on mood and (mental) wellbeing, loneliness can have serious effects on physical health, including greater risk of cardiovascular disease [2] and accelerated physiological decline [3]. Importantly, research shows that being close to, or living in, nature can reduce feelings of loneliness and enhance feelings of social support [4]. Likewise, when people feel related to nature, they experience a greater sense of connectedness to other people and to the world at large [5]. Such findings are in line with the steadily growing body of research showing that interactions with nature are beneficial for (mental) health and wellbeing [6,7].

The following stresses the importance of interacting with nature. As nature becomes ever sparser in urban areas and time spent indoors on screens increases, direct nature experiences become progressively unavailable to new generations [6,8]. In addition, access to nature may be limited due to mobility restrictions (e.g., for older adults) and busy work schedules. Within this context, a promising strategy seeks to complement outdoor nature experience with digital nature experiences generated via immersive technologies (i.e., technological nature—[8]), especially when considering that although simulated nature may not have the same benefits of real nature [9], an ever-growing body of research shows that indirect interactions with nature (e.g., exposure to nature via pictures and videos) can at least confer some of the benefits of real nature experiences [10,11,12]. With the advent of virtual reality (VR) and augmented reality (AR), possibilities for creating ever more immersive and social nature experiences come into play. For instance, findings from a case study show that animated nature projections (aimed at providing relaxation and stimulating social engagement) were successfully implemented at a Dutch care centre, underscoring the potential of ambient technology [13].

However, there is still relatively little research pinpointing key features of nature scenes, let alone guidelines for designing digital nature scenes for enhancing social connectedness. For although the dominant framework in nature research (i.e., attention restoration theory [ART]—[7]) proposes general characteristics which promote restoration in nature such as mystery and coherence (reflecting people’s needs for exploration and control over environmental settings respectively), such features are hard to translate into specific design features of nature scenes (cf. [12]).

Nonetheless, recent research points at the importance of spaciousness in nature which has been related to a social mindset, prosocial behaviour and related emotions such as awe and wonder (e.g., [14]. Arguably, spacious nature scenes (in which one feels small as opposed to something grand) promote social aspirations and prosocial behaviour by reducing self-centredness and promoting a more inclusive sense of self and a broader mindset (i.e., a “collective” rather than a “small” self—[14]). In line with this notion, it has been shown [12] that spacious nature scenes increase creativity and inspiration and are likewise indicative of a broader mindset.

In addition to considering (affective) experiential effects of nature scenes, on a more functional level, it might also be important to consider social affordances of different nature scenes. That is, regardless of spatial landscape characteristics, tended nature scenes might be considered more suitable for social contact than wild nature scenes as they provide more opportunities for social interaction. Finally, we will take into account social context (i.e., by differentiating between individual and social viewing conditions) as it might affect to what extent people are focused on nature and hence “open” to its wholesome effects [15].

Hence, the aim of the current research was to test effects of digital nature projections on social aspirations and related affective states. To this end, an experimental study was conducted in which participants, individually or in pairs, were exposed to immersive projected nature scenes varying in spaciousness (i.e., spacious versus dense) and type of nature (i.e., wild nature versus tended nature). Before elaborating on the details of this study, we will first discuss the key notions involved.

## 2. Theoretical Background

### 2.1. Nature and Social Wellbeing

Nature is an underrecognised healer, a recent report of the Institute for European Environmental Policy says [16]. Indeed, a growing body of research shows that exposure to nature is not just pleasant or aesthetically pleasing but positively influences a wide variety of measures related to wellbeing, including attention restoration [7,11], mood and positive affect [6,12], recovery after medical procedures [17] and creativity and inspiration [12].

Although the influence of nature on diverse facets of wellbeing is well-established, when it comes to social dimensions of wellbeing, research is largely silent. However, findings from several studies hint at the potential of nature to enhance social wellbeing (comprising feelings of loneliness and social support) as well (e.g., [4,18]). For instance, previous research revealed a correlation between the amount of green space in the living area, perceived social support and even loneliness [4]. More specifically, findings suggest that people living near nature generally experience higher levels of social support and less loneliness [4]. These findings suggest that having green space nearby can reframe the perception of loneliness. In line with this notion, another study showed that people with poor social connectedness reported higher levels of wellbeing when they lived nearby nature compared to people with no nature nearby [18].

In line with studies showing that interactions with nature can enhance social connectedness, research zooming in on underlying processes suggests that nature connects people by promoting community-centred goals, whereas urban environments tend to make people feel more selfish by stimulating self-centred goals [19]. Further underscoring nature’s potential to strengthen social bonds, research shows that interactions with nature make people more willing to cooperate [20]. These combined findings highlight the potential of nature as a means to enhance social wellbeing.

### 2.2. Digital Nature Representations

When considering how to make nature accessible for people with limited or no access to nature, studies looking into the effects of (digital) representations of nature are of particular interest. Across a wide range of studies, various types of nature representations have been used (see [11] for a review), varying from real plants or nature elements to images, posters and videos [10,12]. Additionally, nature can also be made accessible via VR or AR, for example as an augmented biking exercise with augmented nature [21,22,23] or virtual nature in nursing homes for recreational purposes [13,24].

As to the effects of different types of nature exposure, research indicates that simulated nature exerts similar benefits as opposed to real nature. For instance, in a comparison of their restorative effects, both real and simulated nature enabled stress recovery, although only the natural environment increased energy and altered states of consciousness, like day-dreaming or imagery [25]. Exposure to a virtual nature environment (with both visuals and sound) had a positive effect on stress recovery [26]. Finally, research shows that virtual nature can elicit specific emotions [27] and sense of presence [28], both of which are not less authentic compared to real-life situations. In short, these combined findings underscore the potential of virtual nature environments to enhance wellbeing.

A key benefit of virtual nature (as opposed to static pictures of nature scenes) is its potential to render nature exposure multimodal. Research suggests that adding matching sounds (i.e., bird sounds are the most commonly mentioned sounds related to restoration, followed by water sounds) enhances immersion and persuasiveness of digital nature scenes [29]. For instance, adding sounds of nature to a virtual nature environment was shown to enhance stress recovery, in contrast to a no-sound condition which did not yield stress recovery [26]. Furthermore, in a study where participants had to compare experiences of natural environments and simulated natural environments, one of the most mentioned discrepancies was smell (or rather the absence of smell in the simulated environment—[25]). In line with such findings, it has been argued that it would be more than worthwhile to use the potential of new technologies by creating multisensory nature experiences [26,30]. However, so far, studies systematically manipulating nature features in (digital) nature scenes to study effects on social experiences are nonexistent.

### 2.3. Variety in Nature Types: Tended versus Wild Nature

Nature is often treated as a unified category, usually pitted against “urban” scenes in research. However, obviously nature comprises many subcategories varying in the extent to which they, amongst others, comprise vegetation, such as trees and plants (and density thereof), the presence or absence of water (such as rivers and lakes) and social or man-made elements (such as benches or paths in gardens or parks).

Nature research generally reveals a preference for wild nature as it is considered more mysterious and hence provides opportunities for exploration and new experiences, an important facet in attention restoration theory [7,31]. Mystery is also associated with attention restoration (ART—[7]), which is often higher in more wild and unpredictable nature [32,33]. Arguably, nature high in mystery is more fascinating (e.g., [12]) and provides a greater sense of “being away”. For example, in a study on wild versus tended nature [34], forest and grassland-based environments were compared to tended nature scenes. Results showed that, overall, the wild forest scene (comprising dense forest vegetation) had the largest restorative effect. In addition, within the tended nature scenes, higher levels of density induced greater restoration [34]. Interestingly, however, the medium-density condition was most preferred by participants, suggesting that restorative value and (aesthetic) preference are not necessarily related.

These combined findings suggest that the more natural an environment is, the more restorative it is. However, when it comes to preference and social wellbeing, it is an open question as to which nature types are most beneficial. Further underscoring this lack of understanding, a recent study found no difference in wellbeing outcomes between simulated park-like grassland and forest-like woodland [35]. Likewise, in a study assessing mental health of local residents as a function of the presence and amount of public green spaces [36], a positive correlation was not only found for parks or other green spaces with an explicit nature focus (i.e., green settings with the specific aim to provide contact with nature) but also for green spaces aimed at recreation or practising sports. Finally, zooming in on social wellbeing, tended nature scenes might be particularly suitable for social interactions as they are readily associated with social presence and hence with safety and security (cf. [15]).

In short, when dealing with nature’s potential to stimulate social aspirations and feelings of connectedness, it is an open question whether wild or tended nature is more appropriate or suitable. Thus, whereas on the one hand, one could argue that wild nature scenes are more fascinating and could function as conversation starters promoting social interaction (cf. [13]), on the other hand, tended nature scenes such as gardens or parks might be considered more suitable as they are readily associated with social presence and safety.

### 2.4. Spaciousness and Social Experience: Feeling Connected

Apart from nature type (i.e., wild or tended nature), when it comes to conceptualising and designing nature scenes, of particular importance are insights on specific (spatial) nature characteristics which define a scene and its experiential qualities. However, there is only limited literature on specific characteristics of natural environments related to (social) behaviour and experience (cf. [6,12]). Nonetheless, both in established frameworks on nature experience [37,38] and in more recent studies (e.g., [12,14]), the importance of spaciousness is acknowledged. Although the experiential effects of spaciousness do not take on a key role in longstanding theories on nature experience, the (evolutionary) importance of, respectively, prospect in a landscape, i.e., being able to see without being seen (prospect-refuge theory—[37]) and extent or scope, which is important to “engage the mind” [38] has been stressed.

Of particular relevance to the current study is research centred on the emotion of “awe”, as it is positively correlated with prosocial behaviour [14]. Awe occurs most often in interactions with “vast” (conceptually similar to “spacious”) nature and is conceptualised as "the feeling of being in the presence of something vast and greater than the self" [39]. Specifically, research suggests that awe diminishes a person’s sense of self, shifting focus away from one’s own concerns (i.e., the “small self”) towards a more collective sense of self (i.e., the “collective self”—[14,40,41]). Indicative of such a collective mindset, participants who experienced awe made more ethical choices afterwards [14] and were more willing to help others [42]. Hence, by shifting attention from one’s daily concerns to the bigger picture in life, awe fosters feelings of connectedness with other people and the world at large (see [43] for a review).

Importantly, nature is a particularly prominent elicitor of awe [41,44]. For instance, in a study where undergraduate students were asked to recall a time when they had experienced a “profound sense of beauty,” the majority of experiences involved vast nature and high levels of awe [44]. In other words, nature exposure can induce awe and inspire prosocial aspirations. Furthermore, considering the importance of vastness, awe should be particularly prominent in spacious, rather than dense, landscapes as these promote opportunities for experiencing oneself as part of a larger whole and for engaging a broader and more inclusive mindset (cf. [12]). By consequence, social aspirations (i.e., the willingness to interact with others) should be higher when immersed in spacious, rather than dense, nature environments.

### 2.5. Nature Experience: Alone or Together?

How does one best enjoy and make the best of interactions with nature? The interplay between company, safety needs and restoration has been studied [15]. Their findings suggest that company might enable restoration by providing safety. However, when there is no need for safety, company thwarts restoration and might lessen attention to nature. In line with these findings, another study [45] showed that people prefer to be alone in nature when they are in need of restoration. These combined results suggest that company is not always beneficial for restoration as it might distract from restorative nature characteristics.

Additionally, literature on therapy and therapeutic landscapes stresses the restorative value of group walk therapy and experiencing nature together by showing that walking together has a positive impact on social interaction [46]. During these walks, participants reported that the social interaction opportunities provided by nature resulted in temporary companionship or "walking-with" others. Findings from a related study [47] showed that walking together with others in nature is beneficial for self-reported mental health but (in line with the foregoing) also suggest that the social interaction required might draw attention away from the natural environment, thereby limiting restoration. Furthermore, in a study on the psychological effects of being alone or together during outdoor physical activity [48], it has been suggested that social nature interaction might interact with the actual environmental setting.

Taking into account the moderating role of being alone or with others on people’s experiences in previous research, we will explore to what extent social context (being alone or with another person) influences nature-related experiences in the current research and whether social context interacts with type of nature. For instance, in a tended nature setting (i.e., a park rather than a forest), safety perceptions are arguably less prominent and hence watching digital projections alone might be preferred. On the other hand, in a wild forest setting, people might prefer being in the company of others (cf. [15]).

Building further on the research discussed, the aim of the experimental study introduced next was to test effects of multimodal digital nature projections (varying in terms of spaciousness and type of nature) on social aspirations, taking into account whether participants watched alone or with another person.

## 3. Method

The study has received approval of the ethical committee.

### 3.1. Experimental Design

The study employed a 2 (spaciousness: dense versus spacious) × 2 (nature type: wild versus tended nature) × 2 (viewing condition: alone or with others, see Figure 1) mixed-design with spaciousness as within-subject variable and nature type and viewing condition as between subject variables, see Figure 2.

### 3.2. Stimulus Development

Using an advanced game development platform (Unity; Gaia Package; https://assetstore.unity.com/packages/tools/terrain/gaia-terrain-scene-generator-42618) four different animated nature scenes (dense/wild nature, dense/tended nature, spacious/wild nature, spacious/tended nature) were developed (see Figure 3 and Appendix A). All scenes comprise a stationary viewpoint and corresponding olfactory (Olfactory stimuli were regulated by an EcoLucht Smart scent machine, https://geurmachine.com/geurmachine/ecoluchtsmart.html. The scent mix used was called “Woodmix”, consisting of a mix of the following ingredients: Cedar wood, Pines, Cypress and Lavender.) and auditory stimuli (i.e., a soundtrack comprising bird sounds). Furthermore, the scenes include animated elements (i.e., movement of trees, vegetation and corresponding shadows). Using two 4K projectors (Projectors were 2 Optoma UHD40 4K UHD beamers), the scenes were projected on a 4 × 2.4 meter blank wall. Both projectors had a resolution of 16 × 9, resulting in a 32 × 9 presentation format (see Figure 1).

#### 3.2.1. Spaciousness Manipulation

As for spaciousness, the nature scene was either dense (comprising a high number of view-restricting trees) or spacious (comprising a limited number of trees, creating the impression of a wide-open landscape), see Figure 3.

#### 3.2.2. Type-of-Nature Manipulation

With respect to nature type (wild versus tended nature): nature scenes either resembled a park environment with man-made objects (including a well-defined path, a bench, a lamppost and flowers) or a wild landscape with none of these objects present (Figure 3).

### 3.3. Measures

All participant responses were recorded using 5-point Likert scales (see Table 1 (Using backward translation methods, we constructed Dutch versions of the original English items. Dutch participants filled in the Dutch version, while participants with other nationalities used the original English items. Table 1 presents the reliability analysis of the combined items. Language was included as a covariate in our analysis.) for an overview).

#### 3.3.1. Social Aspirations

Social aspirations were measured with items tapping the appropriateness of the nature scene for social interaction as indicated by items such as “I would like to show this landscape to someone” and “This landscape is suitable to experience together” (α=0.74).

#### 3.3.2. Awe

Awe was measured with the subconstructs self-diminishment, vastness and connectedness of the original Awe Experience Scale [49], and includes items such as “I felt small compared to everything else” and “I experienced a sense of oneness with all things” (α=0.93). From the 15 items in total, four items were excluded based on a factor analysis, resulting in a final number of 11 items.

#### 3.3.3. Fascination

Fascination was measured with the fascination construct of the original “Perceived Restorativeness Scale” [50]. Example items are: “The setting has fascinating qualities” and “There is much to explore and discover here” (α=0.85).

#### 3.3.4. Sense of Presence

As immersion (e.g., sense of presence) is considered as one of the main benefits of digital (multisensory) nature projections, we included sense of presence in the questionnaire for explorative purposes. Sense of presence was measured with the self-location construct of the original “Spatial Presence Experience Scale” [51] and it taps the extent to which one feels more or less present in the environment presented as indicated by items such as “I felt like I was actually there in the environment of the presentation” and “I had the feeling that I was in the middle of the action rather than merely observing” (α=0.91; see Table 1).

### 3.4. Participants and Procedure

Participants (*n* = 96; 58 females) were recruited on the campus of a Dutch university (age range: 18–35; mean age: 23.48). Most participants had a Dutch (*n* = 54) or German nationality (*n* = 26), others had another European (*n* = 6) or non-European (*n* = 10) nationality. Participants with a Dutch nationality filled out a Dutch version of the questionnaire; international students filled out the original English version. One participant was identified as an outlier (standardised score > 3) and was excluded from the analyses. Upon arrival in the lab, the participants(s) (i.e., one participant in the individual viewing condition and two participants in the social viewing condition; see Figure 1) were welcomed by the experimenter (the first author) who explained the study. Next, participants were asked to fill out an informed consent form. Subsequently, the participants followed the experimenter into the laboratory space where they were seated in front of the nature scenes which were already on display. Participants were randomly assigned to one of the conditions and order of appearance (of the two (high and low) spaciousness scenes) was balanced. Participants were instructed to watch the scenes for approximately one minute before filling out the iPad-administered questionnaire (positioned next to the chair; see Figure 1). Next, the experimenter left the room and entered an adjacent room equipped with a one-way mirror in order to keep track of their progress. Once participants were (both) finished with completing the questionnaire for the first scene, the experimenter switched on the projections for the next scene. After completing the questionnaire for the second scene, a post-questionnaire comprising demographic information and manipulation checks was filled out. Upon completion, the experimenter entered the lab again and thanked the respondents for participation for which they received a 5-euro gift voucher.

## 4. Results

### 4.1. Analyses of Variances

Mixed design ANOVA tests were conducted with respectively social aspirations, awe, fascination and sense of presence as dependent variables, with spaciousness as within-subject manipulation, viewing condition and type of nature as between-subjects factors and Language as covariate. To control for order effects, we additionally included order of appearance as between-subjects factor.

#### 4.1.1. Social Aspirations

There was a significant effect of spaciousness on social aspirations’ scores $\left(F\left(1,86\right)=7,33,p<0.01,partial\eta^{2}=0.08\right)$. Social aspirations scores in the spacious condition (M=3.87,SD=0.75) were higher than in the dense condition (M=3.67,SD=0.55). The main effects of type of nature (F(1,86)<1,p=0.43) and viewing condition (F(1,86)<1,p=0.53) were nonsignificant. This main effect was qualified by a significant spaciousness*viewing condition interaction $F\left(1,86\right)=4,93,p=0.03,\eta^{2}=0.05$, see Figure 4. Pairwise comparisons show a significant (p<0.01) difference in social aspirations’ scores between the spacious and dense variants in the individual condition. There was no such effect (p=0.69) when participants were tested in pairs, showing that spacious scenes were more inviting for social contact in the individual viewing condition only.

All other interaction effects were nonsignificant $\left(p^{\prime }s>0.14\right)$.

#### 4.1.2. Fascination

No significant main effect of spaciousness, type of nature or viewing condition on fascination scores was found $\left(allF^{\prime }s<1,allp^{\prime }s>0.39\right)$. However, there was a significant interaction effect of spaciousness * viewing condition on fascination, $F\left(1,86\right)=9,14,p<0.01,partial\eta^{2}=0.10$, see Figure 5. Pairwise comparisons point at a significant (p<0.01) difference between the spacious and dense conditions in the social viewing condition, suggesting that when participants watched together, the spacious scenes were rated as less fascinating than the dense scenes, while this was not the case when participants watched individually (p=0.32).

All other interaction effects were nonsignificant $\left(p^{\prime }s>0.63\right)$.

#### 4.1.3. Awe

No significant main effects of spaciousness, type of nature (both $F^{\prime }s<1,p^{\prime }s>0.44$) and viewing condition (F(1,86)=1.77,p=0.19) on awe scores were found. There was a significant interaction effect of spaciousness * viewing condition on awe, $F\left(1,86\right)=5,70,p=0.02,partial\eta^{2}=0.06$, see Figure 6. Pairwise comparisons show a significant difference (p=0.04) between the viewing conditions for the spacious condition, suggesting that participants who watched alone experienced more awe in the spacious condition, whereas this difference was not significant for participants who watched together (p=0.66).

No significant interaction effect of spaciousness * type of nature on awe was found (F<1,p=0.32).

#### 4.1.4. Sense of Presence

No significant main effects of spaciousness (F(1,86)<1,p=0.89), type of nature (F(1,86)=1.40,p=0.24) and viewing condition (F(1,86)<1,p=0.39) on sense of presence scores were found. However, there was a significant interaction effects of spaciousness * viewing condition on sense of presence, $F\left(1,86\right)=4,14,p=0.05,partial\eta^{2}=0.05$, see Figure 7. Although pairwise comparisons show no significant difference between conditions, the overall pattern suggests that the sense of presence scores were lower for the spacious condition. However, this was only the case when participants were together.

The interaction between spaciousness and type of nature on sense of presence was nonsignificant (F(1,86)=3.22,p=0.08).

#### 4.1.5. Summary of Mixed ANOVA Effects

Social aspiration scores were significantly higher for the scenes in the spacious (rather than the dense) condition. This main effect was qualified by viewing condition, showing that spacious scenes elicited higher social aspirations’ scores when participants were tested individually only. Additionally, findings show that the spacious scenes were rated as less fascinating than the dense scenes when participants watched together, while this was not the case when participants watched individually. Finally, participants who watched individually experienced more awe in the spacious condition than participants who watched in pairs.

### 4.2. Additional Effects Concerning Order of Appearance

Participants watched two different scenes during the experiment, since spaciousness was manipulated within subjects. Therefore, we also looked into the effects of the order of appearance of the scenes on the most important outcome variable in our study: social aspirations. In the analysis, order of appearance (dense—spacious or spacious—dense) was handled as a between-subjects factor.

#### Effects of Order of Appearance on Social Aspiration Scores

There was a significant interaction effect of spaciousness * order of appearance on social aspiration scores $\left(F\left(1,86\right)=18,77,p<0.001,partial\eta^{2}=0.18\right)$, see Figure 8. Specifically, pairwise comparisons show that in the condition where the spacious scene was followed by the dense scene, the Dense scene was rated lower on social aspirations (p<0.001), while there was no such effect when the dense scene was followed by the spacious scene (p=0.28). Additionally, pairwise comparisons also point out that the dense scene was rated significantly lower on social aspiration scores when it was the second scene, compared to when it was the first scene (p<0.01), while there was no such effect for the spacious scenes (p=0.28).

Additionally, there was a significant interaction effect of order of appearance * viewing condition on social aspiration scores $\left(F\left(1,86\right)=8,03,p<0.01,partial\eta^{2}=0.09\right)$, see Figure 9. Pairwise comparisons show a significant (p<0.01) difference between the different Order of Appearance conditions in the individual viewing condition for social aspiration scores. In the social viewing condition, this difference is nonsignificant (p=0.69. However, in the individual viewing condition, social aspirations were higher when the dense scene preceded the spacious scene.

All other (interaction) effects concerning order of appearance (order of appearance, order of appearance * type of nature, order of appearance * type of nature * viewing condition, order of appearance * spaciousness * type of nature, order of appearance * spaciousness * viewing condition, order of appearance * spaciousness * type of nature * viewing condition) on social aspiration scores were not significant $\left(allp^{\prime }s>0.07\right)$.

## 5. Discussion

Inspired by previous research hinting at the potential of nature to stimulate social aspirations, an exploratory lab experiment was conducted where participants watched digital nature projections (either in pairs or individually), varying in terms of spaciousness (i.e., dense or spacious) and type of nature (i.e., tended nature or wild nature). Most importantly, findings indicate that spacious landscapes enhance social aspirations and the willingness to share (nature) experience, in particular when participants are tested individually.

These combined findings suggest that when the aim is to stimulate social interactions amongst people who are socially withdrawn (i.e., alone) or hesitant to engage in social interaction, spacious nature scenes might be particularly effective. These findings corroborate previous research showing that spacious (indoor) environments stimulate interpersonal communication (i.e., self-disclosure—[52,53]). Additionally, it is worth mentioning that the difference between spacious and dense scenes was particularly prominent when the dense scene appeared after the spacious scene, hinting at the importance of order of appearance and related contrast effects.

In line with [35], social aspirations’ scores did not differ between the tended and the wild nature conditions. It is an open question to what extent this null finding relates to participant demographics such as age and mobility restrictions. For instance, reduced mobility makes access to wild nature troublesome, and objects and environmental design features such as furniture and lighting are perhaps particularly important for old (rather than young) adults with a concern for safety and comfort. On the other hand, it is also true that our tended nature scenes did not depict other individuals or cues indicative of “social presence” in the here and now. Arguably, a lack of social interaction in scenes suggestive of recreation and social contact might accentuate feelings of loneliness, similar to how it is also possible to feel lonely when one is surrounded by a large number of people that one has no actual interaction with [54].

As for fascination, our findings suggest that when participants watched together, the spacious scenes were rated as less fascinating than the dense scenes, whereas this was not the case when participants watched individually. The finding that dense nature is more fascinating is in line with research addressing effects of naturalness on restoration in which density is associated with larger health effects [31]. However, as to why this effect did not transpire when participants watched alone, we can only speculate. Perhaps fascinating qualities of dense landscapes (e.g., more intricate patterns created by a higher number of trees in the scene and overall more stimuli to look at) are still prominent when attention is directed to another person (i.e., in a social viewing condition), whereas more open, spacious landscapes (with less intricate patterns and stimuli to behold) arguably lose more of their fascinating qualities when direction is turned elsewhere. On the other hand, the spatial (“open”) dimension of landscapes has also been pointed out as fascinating [55].

Considering the importance of fascination in attention restoration theory [7], clearly the relationship between specific landscape properties and fascination warrants follow-up research. What is clear from our findings though is that scenes that are most fascinating are not necessarily the same scenes that trigger social aspirations, suggesting that restorative value and social value of nature scenes are not necessarily related. In other words, our results underscore that there is no “one size fits all” solution to nature interventions. Rather, decisions regarding design and selection of nature scenes should be made in light of the specific aims and intentions underlying applications of digital nature for particular people in specific settings. Somewhat surprisingly, type of nature had no effect on fascination scores, perhaps indicating that this manipulation was not explicit enough (especially when considering that the landscapes were, except for the presence of street furniture and a demarcated track, identical).

Likewise, when it comes to the relationship between spaciousness and awe, this study yields interesting findings. Although we found no direct effect of spaciousness on awe (arguably because we used quite everyday types of nature, unlike the grand overwhelming nature scenes used in awe research (e.g., [56]), the interaction between spaciousness and viewing condition indicates that participants who watched individually did experience more awe in the spacious condition. Perhaps also here, the presence of others might distract attention from the positive effects of nature experience on restoration [45]. In other words, being alone might allow the feeling of awe to manifest as it facilitates an engaged or aesthetic viewing mode (rather than an ordinary viewing mode) which is necessary for immersion in nature landscapes and the arts in general [57].

Of further interest is the relationship between awe and prosocial behaviour demonstrated in previous research [14,43] showing that vast (nature) interactions can reduce self-centredness and instead make people feel more connected to others. In line with this notion, a longitudinal study revealed a relationship between self-centredness and loneliness [1]. These combined findings underscore the importance of stimulating awe and related feelings of connectedness with others and the world at large amongst people struggling with loneliness and social isolation. Our findings suggest that spacious nature scenes might be particularly suited for this aim.

### 5.1. Future Research

Future research should point out whether our findings are generalizable to a larger population, specifically for a general ageing population for whom loneliness is a particularly pressing issue. For instance, taking into account the importance of social context revealed in this study, nature projections could be tested across social (care) environments and within living environments of older adults who live independently.

Importantly, such studies should not just include measures for social aspirations and related emotions but should also include more direct measures for loneliness and social isolation. In addition to self-report measures reflecting the extent to which people suffer from loneliness, it would also be important to include observations from caretakers (where appropriate) and family to investigate whether people will actually feel less lonely when interacting with digital nature over longer periods of time.

Additionally, such studies should also shed light on how demographics and mobility constraints shape preferences for different nature scenes. Although no effects of nature type were obtained in this study, participants were students for whom mobility constraints and related safety and comfort issues did not play a role. Hence, in future research the interplay between age, mobility and related safety and comfort concerns should be taken into account by including a larger participant population more representative of our ageing society.

A related issue for future research relates to further unravelling the interplay between safety concerns and social context. For instance, it might be the case that when in the presence of others, one feels safe enough to experience more adventurous stimuli compared to what one would enjoy when watching alone. However, it could also be the other way around, especially when feeling uncomfortable in the presence of others. In those cases, an adventurous scene might not result in a pleasant experience. It would thus be interesting to investigate the interplay between familiarity of the companion (e.g., formal or informal caregivers in a care centre) and type of nature scene.

### 5.2. Limitations

There is still a lack of studies on virtual environments that stimulate additional senses on top of visuals and audio. Our study used audio and olfactory stimuli; however, these were constant across all conditions which prevents us from drawing conclusions on the importance of multisensory stimulation in this context. Likewise, our findings do not warrant firm conclusions on potential matches or mismatches between vision, sound and smell. Hence, in addition to testing different soundscapes and smells, it would also be important to study effects of mismatches between different modalities as these arguably reduce immersion in nature scenes [26]. In addition, sound and scent were constant across the conditions, although one would arguably expect to not just hear bird sounds in the spacious condition but also to see them. (Note that in the dense condition, this is arguably less of an issue as field of vision is limited).

### 5.3. Conclusions

This study shows that digital nature projections can provide exciting new ways to bring nature inside. Although there is no substitute for real nature, the aim of this paper was to investigate whether exposure to digital nature might complement outdoor contact with nature. Our findings suggest that at least some of the benefits of real nature can be conferred through digital means, such that people (regardless of physical status, availability or access to nature) can enjoy beneficial effects of interaction with nature and feel connected to others, even in the face of shrinking networks which old age inevitably brings.

## Figures and Tables

**Figure 1 ijerph-17-01454-f001:**
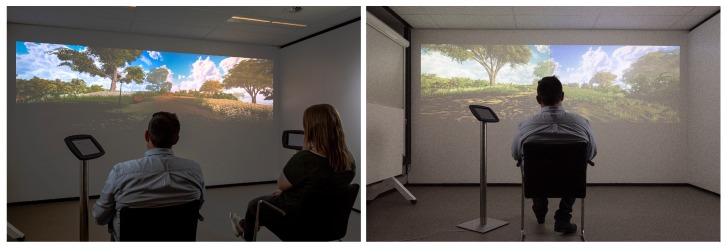
Social (**left panel**) versus individual (**right panel**) viewing condition.

**Figure 2 ijerph-17-01454-f002:**
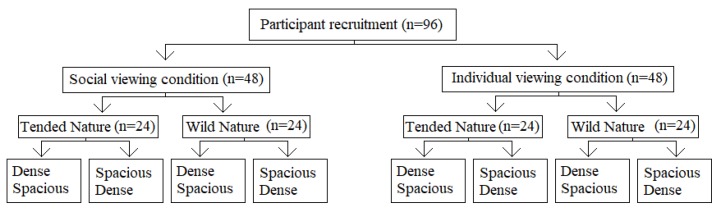
Experimental design.

**Figure 3 ijerph-17-01454-f003:**
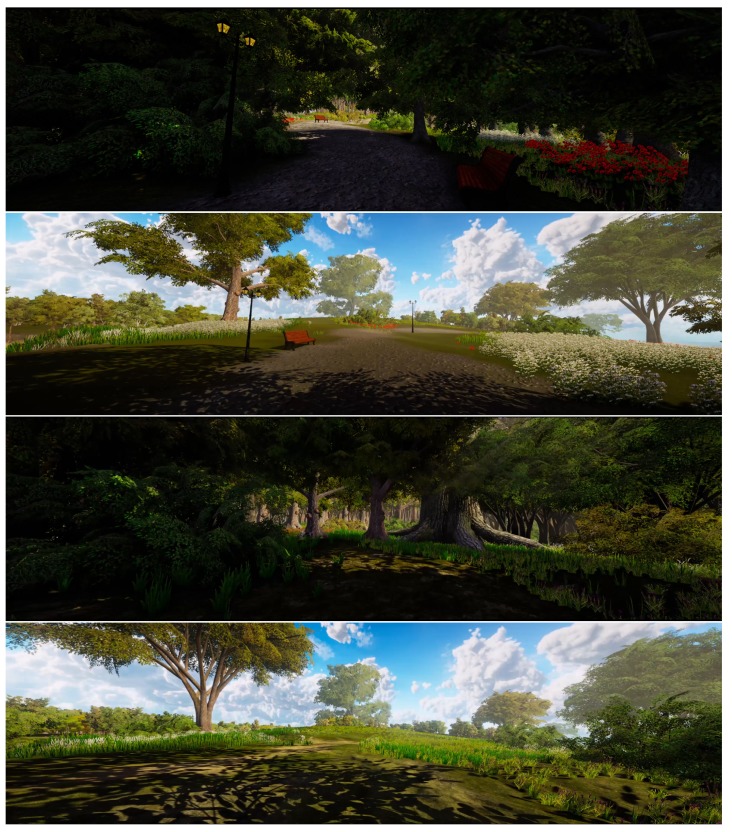
Virtual nature scenes used in main study. From top to bottom: dense-tended nature, spacious tended nature, dense-wild nature, spacious wild nature.

**Figure 4 ijerph-17-01454-f004:**
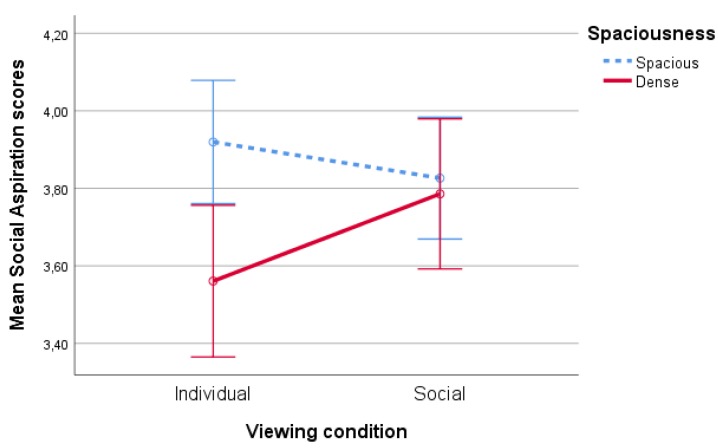
Interaction effect of spaciousness * viewing condition on social aspiration scores.

**Figure 5 ijerph-17-01454-f005:**
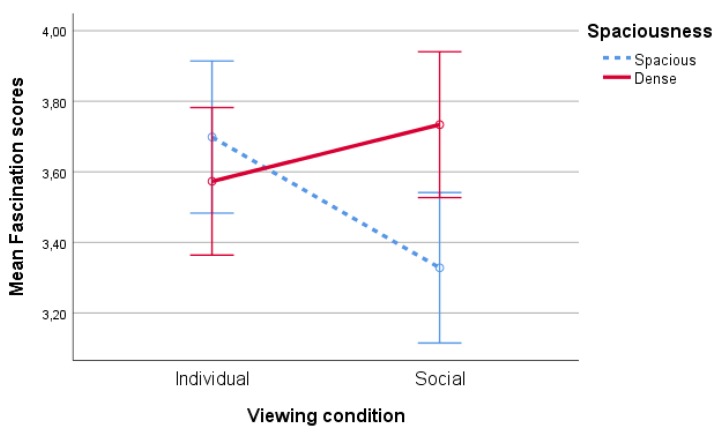
Interaction effect of spaciousness * viewing condition of fascination scores.

**Figure 6 ijerph-17-01454-f006:**
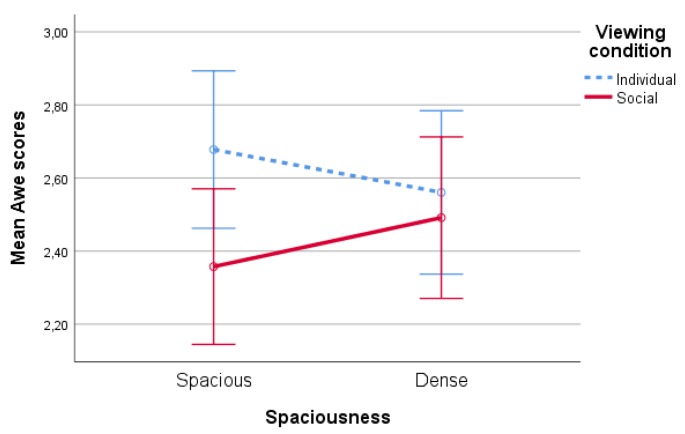
Interaction effect of spaciousness * viewing condition on awe scores.

**Figure 7 ijerph-17-01454-f007:**
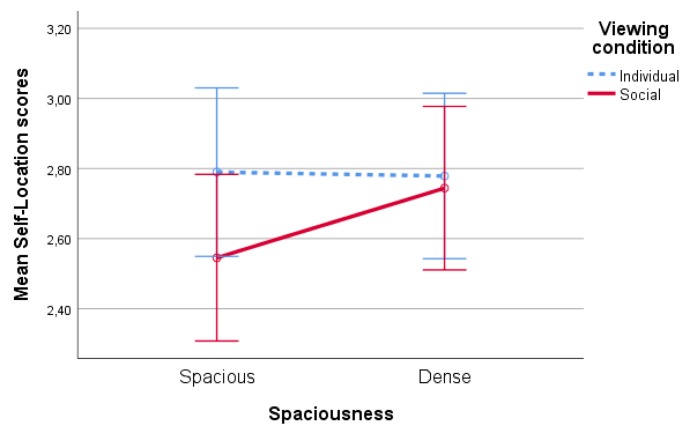
Interaction effect of spaciousness * viewing condition on sense of presence scores.

**Figure 8 ijerph-17-01454-f008:**
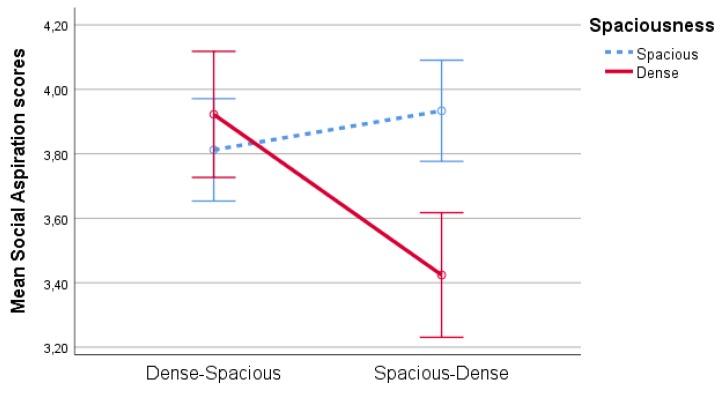
Interaction effect of spaciousness * order of appearance on social aspirations.

**Figure 9 ijerph-17-01454-f009:**
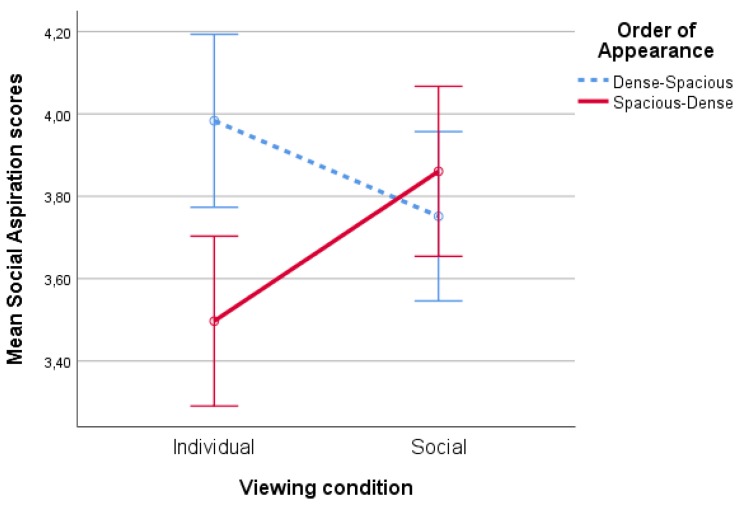
Interaction effect of order of appearance * viewing condition on social aspiration scores.

**Table 1 ijerph-17-01454-t001:** Scales used for measures.

Original Scale	N of Items	α	Items
Social Aspirations Scale (New)	5	0.74	I would like to show this landscape to someone
			I would like to meet here with a friend
			I would like to have a spontaneous chat
			This landscape is suitable to experience together
			If I would encounter someone here, I would feel
			uncomfortable (Reverse)
Awe experience scale [49]	11	0.93	I felt my sense of self become somehow smaller
			I felt small compared to everything else
			I had the sense of being connected to everything
			I felt a sense of communion with all living things
			I experienced a sense of oneness with all things
			I felt closely connected to humanity
			I had a sense of complete connectedness
			I felt that I was in the presence of something grand
			I experienced something greater than myself
			I felt in the presence of greatness
			I perceived something that was much larger than me
Perceived restorativeness Scale [50]	5	0.85	The setting has fascinating qualities
			My attention is drawn to many interesting things
			I would like to get to know this place better
			There is much to explore and discover here
			I would like to spend more time looking at
			the surroundings
The Spatial Presence Experience Scale [51]	8	0.91	I felt like I was actually there in
			the environment of the presentation
			It was as though my true location had shifted into the
			environment in the presentation
			I felt as though I was physically present in the
			environment of the presentation
			I experienced the environment of the presentation
			as though I had stepped into a different place
			I was convinced that things were actually happening
			around me
			I had the feeling that I was in the middle of the
			action rather than merely observing
			I felt like the objects in the presentation surrounded me
			I was convinced that the objects in the presentation
			were located on the various sides of my body

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
