# Peer review of "Does Digital Nature Enhance Social Aspirations? An Experimental Study"

_ijerph, 2020, doi:10.3390/ijerph17041454_

Round 1

Reviewer 1 Report

1) Digital nature may be  useful to address  loneliness or social isolation, but the theoretical mechanism is  indistinct.  

2) The number of  participants is only 96 and especailly 58 females, which results in the conlusions being invalid and  biased. As a possible, I wish it coulde be work to increase more and more participants. 

3) The participants is aged  20-48,   however, they are not only the young and the middle-aged,but also are skilled or a  wide range of interests. Their  loneliness is different form the elderly pepole.  In fact,  loneliness and social isolation is very important challenges of the elderly pepole.  The issue is how to make a distinction between the young and middle-aged and the elderly.

4)  I  am  wonderful that the relationship between spatial loneliness  and digital nature in this study. 

Author Response

First of all, we would like to thank the reviewers for their appreciation of our work. Below we will outline how and where we have addressed the remaining comments and suggestions.

Digital nature may be  useful to address  loneliness or social isolation, but the theoretical mechanism is  indistinct.  

Answer:

In section 2.4 (Spaciousness and Social Experience: Feeling connected), we elaborate on the theoretical mechanism underlying effects of spaciousness in nature (note that for nature type our line of reasoning is more straightforward as we reason that tended nature influences evaluations because of its associations with comfort and safety).

When it comes to spaciousness, our line of reasoning is primarily based on 1) awe research showing that spaciousness in nature stimulates a more collective mindset and triggers awe, and 2) studies confirming the relationship between spaciousness and a collective mindset by showing that people are more willing to engage with others (i.e., have higher social aspirations) in spacious environments (e.g., Okken, Van Rompay, & Pruyn, 2013).

we have now rephrased the final paragraph of 2.4 as follows:

“…considering the importance of vastness, awe should be particularly prominent in spacious, rather than dense, landscapes as these promote opportunities for experiencing oneself as part of a larger whole and for engaging a broader and more inclusive mind set (cf. [10]). By consequence, social aspirations (i.e., the willingness to interact with others) should be higher when immersed in spacious, rather than dense, nature environments.”

The number of  participants is only 96 and especailly 58 females, which results in the conlusions being invalid and  biased. As a possible, I wish it coulde be work to increase more and more participants. 

Answer:

We agree that sample size is limited. However, considering that we employed a mixed-method design with spaciousness as within-subjects factor (meaning that each participant evaluated both a spacious and dense environment) and type of nature as between-subjects factor (participants wither viewed spacious and dense tended nature or spacious and dense wild nature) we considered sample size sufficient. In line with suggestions from reviewer 2, however, we performed a statistical power analysis for sample size estimation with G*Power 3.1 software. With an alpha = .05, power = 0.95 and effect size = .25, the projected sample size needed was approximately N = 96 for this simplest between/within group comparison. Thus, our sample size of N=96 should be adequate for the main objective of this study.

On a more practical note, we would like to point out that sample size was limited as we used a fully immersive lab setup (with large projection screens and complementary multimodal elements [sound and scent]), which we could not transport to other locations in order to recruit larger numbers of participants.

The participants is aged  20-48,   however, they are not only the young and the middle-aged,but also are skilled or a  wide range of interests. Their  loneliness is different form the elderly pepole.  In fact,  loneliness and social isolation is very important challenges of the elderly pepole.  The issue is how to make a distinction between the young and middle-aged and the elderly.

Please note that the participants were aged 18-35. However, we fully agree with the reviewer here and hence in our discussion section (see 5.1. Future Research), we rephrased our take on this issue as follows:

Future research should point out whether our findings are generalizable to a larger population, specifically for a general ageing population for whom loneliness is a particularly pressing issue. For instance, also taking into account the importance of social context revealed in this study, nature projections could be tested across social (care) environments and within living environments of older adults who live independently. Such studies should also shed light on how demographics and mobility constraints shape preferences for different nature scenes. Although no effects of nature type were obtained in this study, participants were students for whom mobility constraints and related safety and comfort issues do not play a role. Hence, in future research the interplay between age, mobility, and related safety and comfort concerns should be taken into account by including a larger participant population more representative of our ageing society.

I  am  wonderful that the relationship between spatial loneliness  and digital nature in this study. 

We agree with the reviewer that the awe and social aspiration measures used in our study do not necessarily map directly onto loneliness and social isolation. That is, 1) feeling part of a collective and 2) experiencing a higher willingness to interact with others does not necessarily entail that one will feel less lonely (although we do feel that these are important antecedents of experienced loneliness). We now elaborate on this issue in 5.1 (Future research) as follows:

“Future research should point out whether our findings are generalizable to a larger population, specifically for a general ageing population for whom loneliness is a particularly pressing issue. For instance, taking into account the importance of social context revealed in this study, nature projections could be tested across social (care) environments and within living environments of older adults who live independently.

Importantly, such studies should not just include measures for social aspirations and related emotions but should also include more direct measures for loneliness and social isolation. In addition to self-report measures reflecting the extent to which people suffer from loneliness, it would also be important to include observations from caretakers (where appropriate) and family to investigate whether people will actually feel less lonely when interacting with digital nature over longer periods of time…”

Reviewer 2 Report

Intro –

The work of Peter Kahn is important to mention. A review of Kahn’s work would suggest that although digital nature has an effect, it is not as good as the real thing. It’s not that the effects are necessarily qualitatively different, just that real nature has a stronger effect.  This is not to undercut the basis or rationale for the current study, but I believe that it is important to cite this work and rephrase the statement “Within this context, it is  particularly interesting to look into possibilities for bringing nature inside, especially when considering  the ever-growing body of research showing that indirect interactions with nature (e.g., exposure to nature via pictures and videos) can yield similar effects as being in real nature [8–10]” so as not to go beyond the data or to suggest that we shouldn’t invest in real nature because artificial nature has the same effect. 

The authors offer concise explanation of the broader mindset and emotions that can be induced through spacious nature as well as the affordances idea of tended green spaces that would explain their specific hypotheses of nature’s impact on their DVs. This was helpful as context and clarity before going into depth in the lit review and helped make the lit review more digestible.

Embedded in the authors’ discussion is the idea that we need to be asking questions about what kind of nature is good for what purpose and for whom. I think it would be helpful to explicitly state this, partly to counter calls for research that ask simplistic questions or draw simplistic conclusions about “how much nature is enough?”  I appreciated that the Discussion section returned to this idea - “It is clear from our findings though is that scenes that are most fascinating are not necessarily the same scenes that trigger social aspirations, suggesting that restorative value and social value of nature scenes are not necessarily related.”

Methods
This is a solid experimental design. It is unfortunate that the control group is considerable smaller than the experimental group. A larger control group would have increased power.

Results:

Results are stated clearly. Given the mixed results in terms of significance, it would be helpful to have a power analysis reported so that we can determine if nonsignificant results are possibly the result of low power vs. likely to be valid.

Discussion:

I have some concern that the following statement is overinterpreting or reaching beyond the data as type of person (lonely for example) was not tested here:  “..when the aim is to stimulate social interactions amongst  people who are socially withdrawn (i.e., alone) or hesitant to engage in social interaction, spacious nature scenes might be particularly effective.”

The authors focus on the implications of this research for older individuals at risk for loneliness. However, the relationship between digital nature and the various dependent variables of interest may be different in that population than the participant sample – young adults. Though the authors address the potential for future research to determine if findings generalize to a larger population, and particularly older individuals, a more direct statement acknowledging that the study is limited in making claims about relevance to older adults (or lonely people as well) should be included.

Overall –

This is a well-written paper that asks an important nuanced question and uses strong experimental methods. A power analysis, however, is missing and I believe is necessary to accurately interpret null findings.  The authors are offering a study that begins to get at an important question at this stage of development of this area of research. We cannot continue to ask “does nature have an impact?” and need to ask “what kinds of nature exposure have what kind of impact, on whom, under what circumstances.” The authors could be of service to the field by calling out this need, and how their study is an attempt to offer this nuanced look at nature’s impact.”

My primary concern about this paper is a tendency to go beyond the data – first in interpreting the literature on the relative impact of digital nature vs. real nature (see Peter Kahn’s body of work) and then in the authors attempt to offer their results as evidence for the benefit of older individuals, older people, and those unable to access real nature. Given that these populations were not the ones studied, I believe this statement of generalizability is overstepping and needs to be toned down somewhat.

Although it is certainly very important that we know how to help those who cannot access nature outdoors to experience nature digitally, I would not want this study’s results to be interpreted as a reason not to work to preserve nature outdoors. I would appreciate some sort of statement reinforcing the importance of quality nature experiences outdoors, and digital nature as an important substitute for some populations when “real nature” is not accessible.

Author Response

First of all, we would like to thank the reviewers for their appreciation of our work. Below we will outline how and where we have addressed the remaining comments and suggestions.

Intro – The work of Peter Kahn is important to mention. A review of Kahn’s work would suggest that although digital nature has an effect, it is not as good as the real thing. It’s not that the effects are necessarily qualitatively different, just that real nature has a stronger effect.  This is not to undercut the basis or rationale for the current study, but I believe that it is important to cite this work and rephrase the statement “Within this context, it is  particularly interesting to look into possibilities for bringing nature inside, especially when considering  the ever-growing body of research showing that indirect interactions with nature (e.g., exposure to nature via pictures and videos) can yield similar effects as being in real nature [8–10]” so as not to go beyond the data or to suggest that we shouldn’t invest in real nature because artificial nature has the same effect. 

Answer:

Thanks so much for pointing out the work of Peter Kahn as we fully agree that his work is very relevant to our work and hence important to mention. In line with Kahn (which we now cite; Kahn, 2011; Kahn, Severson, & Ruckert, 2009), we argue that interacting with digital nature provides some of the benefits but certainly not all of the benefits of real nature. In the introduction section (first paragraph), we now rephrased our argument as follows:

“…However, as nature becomes ever sparser in urban areas and time spent indoors on screens increases, direct nature experiences become progressively unavailable to new generations (Kahn, 2011; Bratman et al., 2019). In addition, access to nature may be limited due to mobility restrictions (e.g., for older adults) and busy work schedules. Within this context, a promising strategy seeks to complement outdoor nature experience with digital nature experiences generated via immersive technologies (i.e., technological nature; Kahn, 2011), especially when considering that although simulated nature may not have the same benefits of real nature (Kahn, Severson, & Ruckert, 2009), an ever-growing body of research showing that indirect interactions with nature (e.g., exposure to nature via pictures and videos) can at least confer some of the benefits of real nature experiences [8–10].”

References:

Kahn, P.H. (2011). Technological nature: Adaptation and the future of human life. London: MIT Press.

Kahn, P. H., Severson, R. L., & Ruckert, J. H. (2009). The human relation with nature and technological nature. Current Directions in Psychological Science, 18(1), 37–42.

The authors offer concise explanation of the broader mindset and emotions that can be induced through spacious nature as well as the affordances idea of tended green spaces that would explain their specific hypotheses of nature’s impact on their DVs. This was helpful as context and clarity before going into depth in the lit review and helped make the lit review more digestible.

 Answer:

Thanks for your appreciation and kind words.

Embedded in the authors’ discussion is the idea that we need to be asking questions about what kind of nature is good for what purpose and for whom. I think it would be helpful to explicitly state this, partly to counter calls for research that ask simplistic questions or draw simplistic conclusions about “how much nature is enough?”  I appreciated that the Discussion section returned to this idea - “It is clear from our findings though is that scenes that are most fascinating are not necessarily the same scenes that trigger social aspirations, suggesting that restorative value and social value of nature scenes are not necessarily related.”

 Answer:

Yes, we fully agree here and now made stress this issue even clearer as suggested by the reviewer (discussion section, 5th paragraph):

“…Considering the importance of fascination in attention restoration theory [7], clearly the relationship between specific landscape properties and fascination warrants follow-up research. What is clear from our findings though is that scenes that are most fascinating are not necessarily the same scenes that trigger social aspirations, suggesting that restorative value and social value of nature scenes are not necessarily related. In other words, our results underscore that there is no ‘one size fits all’ solution to nature interventions. Rather, decisions regarding design and selection of nature scenes should be made in light of the specific aims and intentions underlying applications of digital nature for particular people in specific settings …”

This is a solid experimental design. It is unfortunate that the control group is considerable smaller than the experimental group. A larger control group would have increased power.

Answer:

Please note that we had the same number of participants in all 8 conditions of our experimental design (see Figure 1). However, we do agree that a larger overall sample size would have been desirable (as also pointed out by reviewer 1).

Results: Results are stated clearly. Given the mixed results in terms of significance, it would be helpful to have a power analysis reported so that we can determine if nonsignificant results are possibly the result of low power vs. likely to be valid.

Answer:

We agree that this would be helpful. Hence (also in response to reviewer 1), we performed a statistical power analysis for sample size estimation with G*Power 3.1 software. With an alpha = .05, power = 0.95 and effect size = .25, the projected sample size needed was approximately N = 96 for this simplest between/within group comparison. Thus, our sample size of N=96 should be adequate for the main objective of this study.

Discussion: I have some concern that the following statement is overinterpreting or reaching beyond the data as type of person (lonely for example) was not tested here:  “..when the aim is to stimulate social interactions amongst  people who are socially withdrawn (i.e., alone) or hesitant to engage in social interaction, spacious nature scenes might be particularly effective.”

The authors focus on the implications of this research for older individuals at risk for loneliness. However, the relationship between digital nature and the various dependent variables of interest may be different in that population than the participant sample – young adults. Though the authors address the potential for future research to determine if findings generalize to a larger population, and particularly older individuals, a more direct statement acknowledging that the study is limited in making claims about relevance to older adults (or lonely people as well) should be included.

 Answer:

We agree with the reviewer that our statement “..when the aim is to stimulate social interactions amongst  people who are socially withdrawn (i.e., alone) or hesitant to engage in social interaction, spacious nature scenes might be particularly effective.” is overreaching as we indeed did not test amongst lonely elderly to begin with.

In addition, the awe and social aspiration measures used in our study do not necessarily map directly onto loneliness and social isolation. That is, 1) feeling part of a collective and 2) experiencing a higher willingness to interact with others does not necessarily entail that one will feel less lonely (although we do feel that these are important antecedents of experienced loneliness). We now elaborate on this issue in 5.1 (Future research) as follows:

“Future research should point out whether our findings are generalizable to a larger population, specifically for a general ageing population for whom loneliness is a particularly pressing issue. For instance, taking into account the importance of social context revealed in this study, nature projections could be tested across social (care) environments and within living environments of older adults who live independently.

Importantly, such studies should not just include measures for social aspirations and related emotions but should also include more direct measures for loneliness and social isolation. In addition to self-report measures reflecting the extent to which people suffer from loneliness, it would also be important to include observations from caretakers (where appropriate) and family to investigate whether people will actually feel less lonely when interacting with digital nature over longer periods of time…”

Overall – This is a well-written paper that asks an important nuanced question and uses strong experimental methods. A power analysis, however, is missing and I believe is necessary to accurately interpret null findings.  The authors are offering a study that begins to get at an important question at this stage of development of this area of research. We cannot continue to ask “does nature have an impact?” and need to ask “what kinds of nature exposure have what kind of impact, on whom, under what circumstances.” The authors could be of service to the field by calling out this need, and how their study is an attempt to offer this nuanced look at nature’s impact.”

My primary concern about this paper is a tendency to go beyond the data – first in interpreting the literature on the relative impact of digital nature vs. real nature (see Peter Kahn’s body of work) and then in the authors attempt to offer their results as evidence for the benefit of older individuals, older people, and those unable to access real nature. Given that these populations were not the ones studied, I believe this statement of generalizability is overstepping and needs to be toned down somewhat.

Although it is certainly very important that we know how to help those who cannot access nature outdoors to experience nature digitally, I would not want this study’s results to be interpreted as a reason not to work to preserve nature outdoors. I would appreciate some sort of statement reinforcing the importance of quality nature experiences outdoors, and digital nature as an important substitute for some populations when “real nature” is not accessible.

Answer:

Thanks again for your appreciation, support, and pointing out of relevant literature. We are confident that we have addressed all of these concerns and feel that our current manuscript is now more nuanced. To reiterate our point made in the introduction section that we consider our approach foremost as a “…promising strategy seeks to complement outdoor nature experience with digital nature experiences generated via immersive technologies (i.e., technological nature; Kahn, 2011)”. we now close our manuscript (5.3. Conclusions) as follows:

“…This study shows that digital nature projections can provide exciting new ways to bring nature inside. Although there is no substitute for real nature, the aim of this paper was to investigate whether exposure to digital nature might complement outdoor contact with nature. Our findings suggest that at least some of the benefits of real nature can be conferred through digital means, such that people (regardless of physical status, availability or access to nature) can enjoy beneficial effects of interaction with nature and feel connected to others, even in the face of shrinking networks which old age inevitably brings.”

Round 2

Reviewer 1 Report

Thank the authors for their revision. In my opinion, the research is worth affirmation so that I agree to accept in present form.